# Indwelling Device-Associated Biofilms in Critically Ill Cancer Patients—Study Protocol

**DOI:** 10.3390/pathogens10030306

**Published:** 2021-03-06

**Authors:** Olguta Lungu, Ioana Grigoras, Olivia Simona Dorneanu, Catalina Lunca, Teodora Vremera, Stefania Brandusa Copacianu, Iuliu Ivanov, Luminita Smaranda Iancu

**Affiliations:** 1Anaesthesia and Intensive Care Department, “Grigore T. Popa” University of Medicine and Pharmacy, 16 University Street, 700115 Iasi, Romania; olguta.lungu@umfiasi.ro (O.L.); ioana.grigoras@umfiasi.ro (I.G.); 2Anaesthesia and Intensive Care Department, Regional Institute of Oncology, 2-4 General Henri Mathias Berthelot Street, 700483 Iasi, Romania; 3Microbiology Department, “Grigore T. Popa” University of Medicine and Pharmacy, 16 University Street, 700115 Iasi, Romania; catalina.lunca@umfiasi.ro (C.L.); teodora.vremera@umfiasi.ro (T.V.); luminita.iancu@umfiasi.ro (L.S.I.); 4Laboratory Medicine Department, Regional Institute of Oncology, 2-4 General Henri Mathias Berthelot Street, 700483 Iasi, Romania; copacianubrandusa@yahoo.com; 5Molecular Biology Laboratory, Regional Institute of Oncology, 2-4 General Henri Mathias Berthelot Street, 700483 Iasi, Romania; iuliuic@gmail.com

**Keywords:** biofilm, next generation sequencing, sepsis, infection, cancer, critical illness, vascular catheter, endotracheal tube, urinary catheter

## Abstract

Health care-associated infections are a leading cause of inpatient complications. Rapid pathogen detection/identification is a major challenge in sepsis management that highly influences the successful outcome. The current standard of microorganism identification relies on bacterial growth in culture, which has several limitations. Gene sequencing research has developed culture-independent techniques for microorganism identification, with the aim to improve etiological diagnosis and, therefore, to change sepsis outcome. A prospective, observational, non-interventional, single-center study was designed that assesses biofilm-associated pathogens in a specific subpopulation of septic critically ill cancer patients. Indwelling device samples will be collected in septic patients at the moment of the removal of the arterial catheter, central venous catheter, endotracheal tube and urinary catheter. Concomitantly, clinical data regarding 4 sites (nasal, pharyngeal, rectal and skin) of pathogen colonization at the time of hospital/intensive care admission will be collected. The present study aims to offer new insights into biofilm-associated infections and to evaluate the infection caused by catheter-specific and patient-specific biofilm-associated pathogens in association with the extent of colonization. The analysis relies on the two following detection/identification techniques: standard microbiological method and next generation sequencing (NGS). Retrospectively, the study will estimate the clinical value of the NGS-based detection and its virtual potential in changing patient management and outcome, notably in the subjects with missing sepsis source or lack of response to anti-infective treatment.

## 1. Introduction

Epidemiological data from the last three decades have shown that the incidence of sepsis has increased considerably. Its mortality, readmission rate and length of hospital stay (LOS) are higher compared with those of other causes of hospitalization. The costs required for the treatment of sepsis exceed those needed to rehabilitate patients with heart failure or acute myocardial infarction [1]. Infections in immune-compromised critically ill patients result in reduced chances (up to 50%) of patient survival [2]. The risk of sepsis/septic shock for neutropenic patients ranges between 7 and 45% [3]. Health care-associated infections (HCAIs) are a leading cause of inpatient complications and 7-10 out of 100 hospitalized patients acquire HCAIs [4]. The most common HCAIs are catheter-associated urinary tract infections (world’s most common HCAIs), ventilator-associated pneumonia (second cause of HCAI in the Intensive Care), postoperative wound infections, gastrointestinal infections and central line-associated bloodstream infections [4]. In modern intensive care, the use of indwelling medical devices is the standard of care. The number, type, exposure time and frequency of insertion may increase according to case complexity. The inserted ID exceeds the body’s physical protective barrier is recognized as non-self and may promote microorganism adhesion leading to colonization, biofilm formation and finally infection. ID-associated HCAIs increase in-hospital morbidity, mortality and costs. 

Microorganisms can exist in planktonic form, which involves freely suspended individual cells in a liquid medium or in the form of a sessile community in biofilms [5,6,7,8,9]. A microbial biofilm represents a three-dimensional structure and is composed of aggregates of microbial cells surrounded by a self-produced polymer matrix, which has a protective role against the hostile environment [6]. Biofilm formation is a dynamic process that includes the following stages: adhesion/attachment, aggregation, maturation, mature biofilm and dispersion [9]. 

These microbial populations defended by the biofilm matrix are adherent to each other, to inert surfaces or to living tissues [7,9]. Following attachment, the formation of the biofilm takes place with the help of a complex mechanism called “quorum sensing”: advanced cell-to-cell chemical communication by secretion of signaling molecules named autoinducers [10,11]. The initiation of quorum sensing takes place only if a sufficient number of microorganisms are present, which increases with raising fluid flow rate [10,12]. Different pathogens exhibit diverse formation behaviors, depending on their quorum sensing-states: *Pseudomonas aeruginosa* forms biofilms at high-cell density, while *Staphylococcus aureus* prefers a low-cell density environment [10,11]. The concentration of signal molecules is also dependent on the cell’s position in the biofilm, matrix’s characteristics and biofilm’s thickness [11]. At first, quorum sensing is heterogeneous, while in mature biofilms autoinducers production and signaling is homogenous [10]. The release of low-molecular weight molecules can initiate cell density-dependent control of gene expression for a particular phenotype of increased or changed virulence [10,12,13]. Subsequently, sessile forms of microorganisms covering the biofilm can give rise to planktonic structures, which can disperse into the body and generate new biofilms [7].

Therefore, microorganisms can survive in a dormant state in these protected communities, which are distinguished by slower metabolism and decreased sensibility to the effect of antimicrobial treatments [5]. A mixture of innate and induced mechanisms are involved in biofilm antibiotic resistance [13]. These include delayed or failed penetration of antimicrobial, horizontal gene transfer (*Pseudomonas aeruginosa*), multidrug efflux pumps, interactions between bacteria and fungi, due to a fungal matrix barrier [12]. The phenotype of the biofilm cells is different from that of planktonic cells. Biofilm cells are unable to grow and produce colonies once extracted from the biofilm community and placed on culture media due to loss of quorum sensing [14]. 

Despite the impressive technological advancement and relevant studies performed in this scientific field, the basic treatment of sepsis has not been altered considerably and the inpatient mortality of the septic patients remains unacceptably high, between 13.6 and 39.3% [15]. The most frequent etiological agents are the Gram negative bacteria (*Escherichia coli*, *Klebsiella* spp., *Pseudomonas* spp., *Acinetobacter* spp.), but recent studies show an increased incidence of Gram positive and fungal induced sepsis [15]. Despite a decreasing incidence in high-income countries, ventilator-associated pneumonia, central-line-associated blood stream infections and catheter-related urinary tract infection are still the most frequent cause of device-related sepsis mortality [16,17]. The latest sepsis guidelines continue to recommend the early approach “Hour-1 bundle” (within the first hour of sepsis suspicion/diagnosis), including early initiation of antibiotic therapy [18,19].

Major challenges in the improvement of sepsis management include: having the shortest delay possible for treatment achievement and utilizing the most accurate method for pathogen detection/identification. However, microorganisms have the ability to change their growth characteristics from an acute attack by planktonic cells, to a slow, prolonged, protected growth, which evades the classical methods of detection [14]. 

The current standard of microorganism identification relies on cell culture growth. The process is time-consuming with low sensitivity and often involves intensive work. Therefore, clinicians initiate empiric broad-spectrum antimicrobials at an early stage, which may imply several risks, such as inappropriate regimen dose, toxicity, *Clostridioides* (former *Clostridium*) *difficile* infections, allergic reactions, multidrug-resistant (MDR) microorganism selection and increased hospital LOS and costs [20,21]. Taking into account that the US National Institute of Health and the Centers for Disease Control and Prevention estimate that 65% and 80% of infections, respectively, are caused by biofilms [22] and that biofilm-associated infections are more difficult to detect by conventional methods, a high number of patients are at risk of being misdiagnosed.

The ideal diagnostic test for sepsis should meet numerous criteria as follows: rapidity in microorganism identification (less than 3 h), ease to perform with minimal technical expertise, ability to detect any type of pathogen (bacteria, fungi, viruses) including new emerging microorganisms, minimal invasiveness, a small volume specimen, high sensitivity and specificity and the ability to differentiate pathogens from contaminants, to assess anti-infective drug resistance and to distinguish between host-derived and pathogen-derived inflammation [21]. 

Standard culture methods exhibit several limitations. The results can be obtained within a highly variable timeframe: 6-8 hours for fast-growing bacteria, 5 days for the majority of organisms including the HACEK group of fastidious bacteria and *Brucella* spp. and more for slow-growing bacteria, fungi and *Mycobacterium* spp. [23,24,25]. This time is required for microorganism growth to detectable levels and for pathogen identification and antimicrobial susceptibility testing (AST). The number of microorganisms available for cultivation is usually small and repeated sampling is recommended for blood cultures, leading to the need of larger samples [25]. During empirical antimicrobial treatment, pathogen identification may be false negative [21]. Deficient antiseptic procedure during sample harvesting can lead to false positive results, leading to pseudobateremia (due to contaminants) [26]. 41–50% of false positive blood cultures (contaminated mostly with coagulase-negative staphylococci, but possible with multidrug resistant strains [26,27]) are likely to be treated with antimicrobials [28], which leads to increased expense and antibiotic resistance [29].

During the last decades, the advances in gene sequencing research have enabled the development of culture-independent techniques for microorganism identification. These methods aim to improve the etiological diagnosis and the antibiotic treatment in order to change sepsis outcomes and to promote antibiotic stewardship policies [30,31,32,33]. Gene sequencing techniques are able to detect fastidious, difficult to grow, anaerobic or dormant microorganisms [32,34]. Despite their high current costs and expertise requirement, rapid technological development and workflow standardization [35] promote large-scale use, while ongoing research aims to establish their usefulness in clinical practice. Although the novel molecular methods have improved detection of biofilm microorganisms, they lack the ability to distinguish between colonization/contamination and infection. A growing number of studies aimed to translate into guidelines the applications of the Next Generation Sequencing (NGS) method to different microbiological products (blood, sputum, urine) and develop validated thresholds [36,37,38]. In the past 10 years, several studies were published on the usage of unbiased metagenomics for the identification of pathogens in biological samples based on Illumina sequencing [35,39]. 

Cancer patients are prone to develop healthcare-associated infections due to the following reasons: acute and chronic immune system dysfunction due to anticancer and antineoplastic treatments (surgery, chemo-, radio-, immune therapy, corticoids, bone marrow transplant), repeated hospitalization and cancer- and treatment-associated malnutrition. The presence of a critical illness is characterized by serious physiological and metabolic derangements, as well as organ dysfunction that require admission, monitoring, diagnosis and treatment in the intensive care unit (ICU) [40]. ICU specific management and environment add a multitude of infection risk factors: invasive procedures, indwelling devices (IDs), cross-contamination, high use of antibiotics and MDR germ selection. The endotracheal tube (ET), the central venous (CVC), arterial (AC) and urinary (UC) catheters are standard care provided in the ICU. The prolonged exposure to these IDs and their frequent manipulation, along with endogenous factors, promote biofilm formation and HCAIs, which are often of unknown origin. The present study hypothesizes that ID-associated biofilms result in the development of HCAIs in critically ill cancer patients and influence their outcome. It was hypothesized that DNA amplification and gene sequencing techniques are superior to standard culturing methods used in biofilm pathogen identification, mainly due to microbes from biofilms being considerably difficult to cultivate.

## 2. Expected Results

The present study aims to evaluate the performance (sensitivity, specificity, positive and negative predictive value) of the NGS-based identification technique compared with that of the conventional culture-based method, for the same ID biofilm samples. The investigators will retrospectively estimate the clinical value of the NGS-based detection and its potential in altering patient management and outcomes, notably in those subjects with missing sepsis sources or lack of anti-infective treatment response.

As standard of care, all septic patients with the need of endotracheal intubation have concomitantly inserted a CVC, an AC and a UC. The investigators will compare in each patient the biofilm-associated pathogens from all four collected IDs in order to assess device or patient specificity. The ID biofilm-associated pathogen data will be compared with the results of the nasal, pharyngeal, rectal and skin pathogen screening. In addition, the investigators will compare biofilm-associated pathogen data with the identified etiological agents of diagnosed infections (respiratory tract/urinary tract/bloodstream/surgical site infection).

The investigators will obtain an overview of the four ID-associated (ET, CVC, AC, UC) biofilm pathogens in septic cancer patients. Specifically, pathogens will be categorized according to taxonomy, number of colonies, ability to grow in culture and antimicrobial resistance. Correlations will be made with the ID exposure time, inflammatory markers, immunosuppression risk factors, severity scores, multiple organ dysfunction evolution, ICU/hospital LOS and disease outcome.

## 3. Discussion

The prevention and treatment of HAIs is a main focus of research investigation as stated by the European Society of Clinical Microbiology and Infectious Diseases Biofilms study group [6]. The essential strategies to reduce its consequences are the early identification of infection source and the initiation of appropriate anti-infective therapy. These goals can be achieved only with multidisciplinary approach, including the use of molecular microbiology [6].

The DNA-based detection methods offer the advantage of using non-viable cell-based methods to assess the clinical course of the patient [41]. In order to circumvent the disadvantage of not being able to distinguish between colonization/contamination and infection, investigators developed the Sepsis Indicating Quantifier score for circulating cell-free DNA, which is a future promising statistical tool for this discrimination [30]. The assessment of relevance of the results needs a large control cohort with non-infected patients allowing an unambiguous identification of pathogens that exactly match with cultures from corresponding patient negative or positive samples [30].

Another important point worth of consideration is the evaluation of microbial species, which play an active role in biofilm-associated infections [6]. Highly vulnerable septic cancer patients are diagnosed with conventional methods. However, it may be difficult to identify an etiological agent, and consequently it is urgently required to improve pathogen identification/diagnosis [42].

Our study is conducted in an Oncology Institute (Regional Institute of Oncology Iasi, Romania, a tertiary 300 beds hospital), which manages oncological patients from the whole North Eastern region of Romania (comprising about 25% of the Romanian population). This is a pilot study, which can be used as the foundation for further multicentric studies involving larger cohorts of patients. However, because a single center study is never enough to make bold recommendations, further research will be needed to support the results, before the study’s findings are considered applicable to medical practice. Future studies may also focus on subcategories of patients of specific clinical relevance, such as neutropenic patients.

This study is funded from the budget of a doctoral grant from “Grigore T. Popa” University of Medicine and Pharmacy, Iasi, Romania, so as a starting point in this research direction, a single center approach is more affordable to conduct than a multicenter one. We point out the fact that in each enrolled patient a number of at least 5 microbiological tests will be performed, which means a total of at least 500 microbiological investigations and over 400 molecular biology processed samples.

## 4. Materials and Methods

### 4.1. Objectives

The primary objective includes the detection/identification of biofilm-associated pathogens using next generation sequencing techniques (NGS) compared with standard microbiological diagnosis. 

The secondary objectives include the following: Identification of pathogens involved in ID biofilm formation (ET, CVC, AC, UC) in critically ill cancer patients;Comparison of four IDs derived from biofilm-associated pathogens collected from the same patient;Comparison of the biofilm-associated pathogens with those identified in currently used biological samples (tracheal aspirate/bronchoalveolar lavage, blood culture, urinary culture and surgical wound swab) collected from the same patient;Examination of correlations between different parameters: biofilm-associated pathogens and patient clinical and biological data, such as nasal, pharyngeal, rectal and skin pathogen screening data, will be assessed, as well as risk factors, such as neutropenia, chemo/radiotherapy, corticosteroid treatment and previous anti-infective therapy, ID exposure time and biological markers of inflammation. The diagnosed infection will also be used as an index and includes respiratory tract infection, urinary tract infection, bloodstream infection, surgical site infection and sepsis of unknown origin. Finally, the severity scores will also be assessed as follows: Sequential [Sepsis-Related] Organ Failure Assessment (SOFA) score and Acute Physiology and Chronic Health Evaluation II (APACHE II) score. The ICU and hospital LOS and the patient outcome, which are categorized as survival/death will be also included in the analysis.

### 4.2. Study Design

The present study is a prospective, observational, non-interventional, single center study ongoing since June 2019. The study refers to the Strengthening the Reporting of Observational studies in Epidemiology guidelines.

### 4.3. Settings

The trial is conducted in the ICU of a University hospital dedicated to cancer patients—Regional Institute of Oncology (RIO) Iasi, Romania, which includes Oncology, Surgery, Anesthesia and Intensive Care, Radiotherapy, Hematology and Palliative Care departments. The primary diseases to which the institute provides management are digestive, gynecological, urological, thoracic, skin and soft tissue and ENT cancers, leukemias, lymphomas, myelomas and malignant immunoproliferative diseases. The 11 beds ICU follows the recommendations of the Society of Critical Care Medicine regarding ICU admission and discharge criteria [43]. In this hospital, an ICU rapid response team ensures on-demand hospital ward guidance for the management of patients with deteriorating conditions, as well as the evaluation of patients that require ICU admission.

All collected samples (biological products or devices) are processed in the Microbiology Laboratory of RIO Iasi. Gene sequencing will be performed in the Center for Fundamental Research and Experimental Development in Translational Medicine—TRANSCEND, RIO Iasi. Both institutions are accredited by national authorities according to international standards.

### 4.4. Eligibility Criteria

The investigators will enroll all consecutive surgical and non-surgical critically ill cancer patients admitted to the ICU, who meet all the inclusion criteria and do not comply with any of the exclusion criteria. Patient’s gender will be taken into consideration when analyzing the data, since gender specific differences exists in the incidence and mortality of certain cancers. All adult cancer patients with suspected/proven sepsis/septic shock of any etiology who are classified according to the new sepsis definitions (Sepsis-3) will be eligible for study inclusion [15].

The investigators aim to enroll 150 adult cancer ICU patients. To identify eligible subjects, the study investigators will carry out a daily screening of all ICU admissions. They will further fill out the patient screening record for each excluded patient and will describe the reason for exclusion. While hospitalized in the ICU, all consecutive participants are managed according to the hospital protocols and clinical judgment of the intensivist without any intervention. They will be surveyed starting with the first ICU day until the last study ID will be removed in the hospital. This may occur during/after ICU admission/discharge, or at patient death. 

#### 4.4.1. Inclusion Criteria

The following inclusion criteria will be used:Signed informed consent;Age ≥18 years;Suspected/proven sepsis/septic shock;APACHE II score ≥10;Predictable invasive ventilatory support ≥48 h;Patient estimated survival ≥4 days.

#### 4.4.2. Exclusion Criteria

The following exclusion criteria will be used: Patient/legal representative refusal;Age <18 years;Chronic psychiatric/neurological disease with impaired decision-making capacity;Pregnancy;Invasive ventilatory support <2 days;Death in less than 4 days following ICU admission.

### 4.5. Ethical Aspects

The present study was approved by the Research Ethics Committee of “Grigore T. Popa” University of Medicine and Pharmacy, Iasi, Romania (approval number 2352 on the 7 March 2019) and of the Research Ethics Committees of RIO, Iasi, Romania (approval number 21 on the 11 February 2019).

During the first 24 h of ICU admission, all eligible patients will receive written information regarding the study implementation, aims, expected advantages and possible risks. Subsequently, they will be asked to sign an informed consent. If the patient will be unable to provide consent at ICU admission due to pathological or drug-induced acute alteration of consciousness, a legal representative will provide the authorization. Once the participant regains the ability to take decisions, he/she will be asked to confirm or withdraw his/her consent.

### 4.6. Description of the Used Methods

#### 4.6.1. Standard of Care in the ICU

According to the RIO protocols, all septic patients with the need of invasive ventilatory support have concomitantly inserted a CVC, an AC and a UC, as standard of care. All patients undergo the protocol for the management of suspected/proven sepsis as follows: initial resuscitation, specimen collection for microbiology/molecular biology tests, empirical/targeted anti-infective treatment, source control, multiple organ support and treatment of the underlying disease/comorbidities. All RIO patients are screened for nasal, pharyngeal and rectal pathogen colonization at the time of hospital/ICU admission. 

#### 4.6.2. Swab Sampling

The nasal, pharyngeal and rectal screening swab sampling are performed according to standard methods. The culture media used are the following: Columbia blood agar and Chapman medium (mannitol salt agar) for nasal swabs, Columbia blood agar, MacConkey agar and Sabouraud dextrose agar with chloramphenicol for pharyngeal swabs and MacConkey agar and Chapmann medium. Specific media are used for carbapenemase, an extended spectrum β lactamase, for vancomycin resistant enterococci and for the identification of *Salmonella* spp. used in the rectal swabs. 

In addition to this standard screening, in the first 24 h of ICU admission cutaneous samples from the groin area of enrolled patients will be obtained [44]. The sampling will be performed with sterile Copan eSwab^TM^ swabs, a product recommended for aerobic, anaerobic and fastidious microbial agents. According to the manufacturer’s recommendations, the skin sample will be obtained prior to the assessment of the patient’s general hygiene, with immediate elution in liquid medium for storage and transport. The samples will be transported as soon as possible to the Microbiology Laboratory for processing. The initial inoculation media are the following: Columbia blood agar, MacConkey agar and Sabouraud dextrose agar with chloramphenicol. Subsequently, specific differentiation and identification media will be used. The final step of pathogen and AST evaluation involves specific assays performed by MicroScan Walk Away 40 plus^®^ (Beckmann Coulter, Inc.).

#### 4.6.3. Biofilm Sampling and Transport

The extraction of the four IDs (ET, CVC, AC, UC) will be performed when the clinical condition of the patient dictates it (suspected catheter infection/no further need due to improvement or death). These devices will be extracted by medical ICU personnel, only at the indication and according to the medical judgment of the clinician, without being influenced by the patient’s study participation. 

Investigators define the ID distal end as the internal part, which is the segment inserted in the respective anatomical structure and the ID proximal end as the external part, which is available for manipulation and connected to a medical equipment [45].

Following aseptic extraction, a sterile sampling of each collected ID will be performed as soon as possible as follows: for the CVC and the AC, following iodine tincture skin disinfection and device extraction, the distal part (approximately 4 cm) will be collected, according to routine practice, which is consistent with the general recommendations [46]. This part will be halved with a sterile scissors, in the two following segments: one for standard culture and the other for DNA analysis. 

The ET’s distal part (approximately 3–5 cm) will be collected according to recommended techniques [45,47]. Subsequently, the sample will be divided with a sterile scissors into 4 pieces, 2 of which will be used for microbiological processing and 2 for gene sequencing. For each type of processing, one part includes a portion of the cuff segment (Figure 1). 

The UC sampling involves collection of the distal part (approximately 5 cm). Subsequently, the sample will be divided into 4 discs, which are less than 1 cm thick, 2 of which will be used for microbiological processing and 2 for gene sequencing [48]. Each type of processing includes one part with a portion of the cuff segment (Figure 2). 

The samples will be transported in sterile recipients and will be transferred as soon as possible to the Microbiology Laboratory for processing. The samples for gene sequencing will be transported to the Molecular Biology Laboratory and stored by freezing at −20 °C until processing.

#### 4.6.4. Standard of Care in the Microbiology Laboratory

Microbiological analysis will be performed by standard methods as follows: Initially, sample inoculation will be conducted on standard culture media (Columbia blood agar, Chocolate blood agar, MacConkey agar and Sabouraud dextrose agar with chloramphenicol), followed by biochemical identification and AST assessment according to the CLSI standards and guidelines [49], which will be accomplished using MicroScan Walk Away 40 plus^®^. The Beckmann Coulter automatic system compatible panels will be used for biochemical testing. CVC and AC samples will be processed by the routine protocol according to the semi-quantitative method proposed by Maki et al. [50]. The roll-plate technique will be used. The sample will be rolled around five times on the culture media and the external germs will be inoculated. The colony forming units will be counted after 24 and 48 h of incubation at 37 ºC and a number of 15 or more will be considered significant [46].

#### 4.6.5. Biofilm Microbiological Processing and Analysis

Since there is no standardized method used for the detection and identification of UC and ET biofilms, the investigators will utilize adapted techniques described in previous publications. ET and UC samples will be processed by the roll-plate technique and the rolled fragments will be placed on the medium surface for 48–72 h incubation at 37 °C (Figure 3 and Figure 4) [45,47,48,51]. 

The colonies observed on sample surrounding area (germs from the inner sample surface) and on distant areas (germs from the outer sample surface) will be collected separately and processed by standard methods.

#### 4.6.6. Biofilm NGS Processing and Analysis

Following complete sample collection, gene sequencing of the variable regions V3-V4 16S rRNA gene will be performed using the Illumina MiSeq^®^ NGS system. 

The ID samples will be slowly defrosted and left at room temperature. The silica sphere DNA extraction kit of the FastPrep-24^TM^ MP Biomedicals device will be used for lysis of bacterial and fungal cells. Microbial DNA extraction will be performed with Wizard^®^ Genomic DNA Purification kit and subsequently a fluorimetric quantification of extracted DNA will be performed with the Qubit^TM^ dsDNA HS Assay kit (Thermo Fisher Scientific, Inc., Waltham, MA, USA) according to the manufacturer’s protocol. 

A first amplification will be carried out with specific primers for the variable regions V3 and V4 of the 16S rRNA gene using the KAPA HiFi^TM^ HotStart ReadyMix kit. Preparation of the fragment libraries will be accomplished using pairs of specific adapters for the sequencing. The platform Illumina will be used on a TruSeq Index Plate Fixture. The second stage will involve a new PCR amplification and the obtained PCR products will be purified with the Agencourt^®^ AMPure XP kit. Subsequently, the investigator will conduct fluorometric quantification, normalization, mixing and distortion of fragment libraries, according to the manufacturer’s protocol.

The sequencing will be achieved with the aid of the MiSeq^®^ Reagent v3 kit on the Illumina MiSeq^®^ platform. The resulting data will be subsequently compared with the international BaseSpace databases and statistically processed in order to identify the microbial populations with regard to the genus and species.

### 4.7. Data Collection and Follow-Up

The case report form includes three sections as follows: Patient data, which involves inclusion and exclusion criteria, demographic data (age, gender, weight, height), hospital admission (day, reason), ICU admission (day, type, admission source, primary admission diagnosis), patient chronic illnesses and co-morbid conditions and ICU admission scores (APACHE II, quick SOFA and SOFA), immunosuppression risk factors, ICU/hospital LOS, 28-Day status;Sepsis monitoring, which includes suspected/proven infection criteria, scoring (daily SOFA score), colonization status, microbiological monitoring, anti-infective treatment and source control and 1st, 3rd and 7th ICU day evaluation of sepsis evolution (shock criteria, inflammatory markers, multiple organ dysfunctions, multi-organ support therapy);Monitoring of each ID, which includes insertion date, ID manipulation (endotracheal aspiration, bronchoalveolar lavage, measurement of intra-abdominal pressure, arterial blood/central venous blood/urine sampling), date of extraction, results of biofilm pathogen identification by culture growth and NGS and antimicrobial resistance profile.

### 4.8. Strategies to Ensure Adequate Enrolment and Protocol Compliance

The data will initially be collected on paper-based case report forms and subsequently will be transferred on a secure electronic database. The protocol compliance, the number of enrolled patients and data accuracy will be periodically assessed. An individual checklist with patient specific time schedule will be used to facilitate appropriate data and sample collection and processing. The investigators will actively monitor all patients leaving ICU with at least one ID in place to secure correct data and sample collection.

### 4.9. Statistical Analysis

Statistical analysis will be performed using the IBM SPSS software, which evaluates the NGS performance in comparison with the standard microbiological method, applied to the same biofilm. The investigators will consider a “*p*” value of 0.05 or less statistically significant. The results will be compared using non-parametric and parametric statistics, according to the types of variables analyzed. In order to define patient characteristics, the investigators will also use descriptive statistical methods.

### 4.10. Sample Size Power Calculation

The primary aim of the study is to characterize the accuracy of Next Generation Sequencing indwelling device-associated biofilm microorganism detection in septic cancer patients in comparison with standard culture, which was estimated to be 35–50% better based on existing literature results [52,53]. Using this estimate, 100 patients are enough to detect the increase in incidence with a power of 80% and a confidence level of 95%.

### 4.11. Strengths and Limitations of the Study Protocol

To the best of our knowledge, this is the first prospective observational study, which aims to compare two methods of pathogen detection/identification responsible for ID biofilm formation in critically ill cancer patients. Concomitantly, it is the first study, which aims to characterize biofilms on four concomitantly implanted IDs used in critically ill patients in correlation with the colonization/infection status. The results of our study may contribute to the understanding of biofilm-associated infections in critically ill oncological patients and to the evaluation of the NGS’s diagnostic power in this particular category of patients. By complex biofilms analysis and correlations with the clinical, treatment and outcome data, the study may bring new insights into the approach of bacterial multi-drug resistance in high-risk patients needing invasive procedures.

Despite the fact that cancer patients are a heterogeneous group in terms of localization, stage and treatment methods, they represent a high research priority. The cancer patients who are critically ill exhibit characteristic features, which should be taken into account for individualized management. The complex microbiological context of these patients and their high predisposition to colonization and infection represent common ICU challenges. The investigation of the biofilm formation in IDs may provide clinical information that can improve future patient management.

A major limitation of the present study is its observational, non-interventional nature, taking into account the fact that ID extraction and biofilm detection will be achieved at the end of the medical intervention. It is difficult to evaluate the ID biofilm temporal occurrence and dynamics due to the unpredicted and variable exposure time. Another limitation is the lack of a control group, consisting of non-septic critically ill patients, who benefited the four IDs. The cohort size is limited by the financial burden caused by NGS use. Heterogeneity of the cohort in terms of cancer- and treatment-specific factors and causes of immunosuppression may challenge generalization of result interpretation. 

## Figures and Tables

**Figure 1 pathogens-10-00306-f001:**
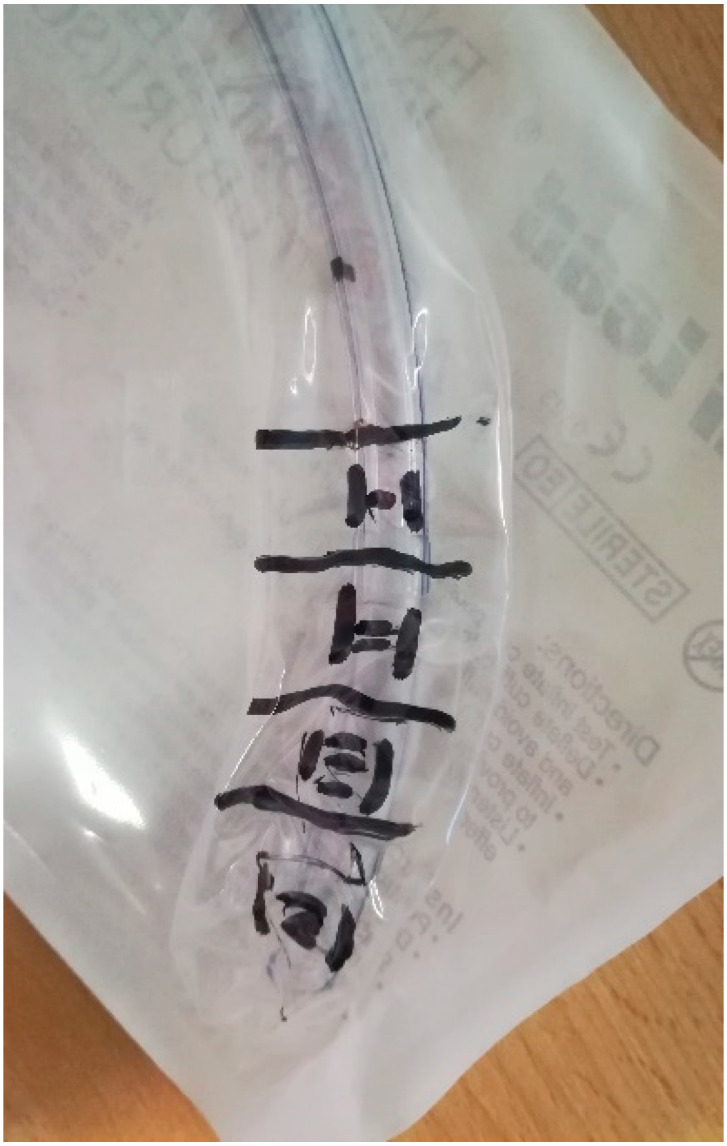
ET sampling.

**Figure 2 pathogens-10-00306-f002:**
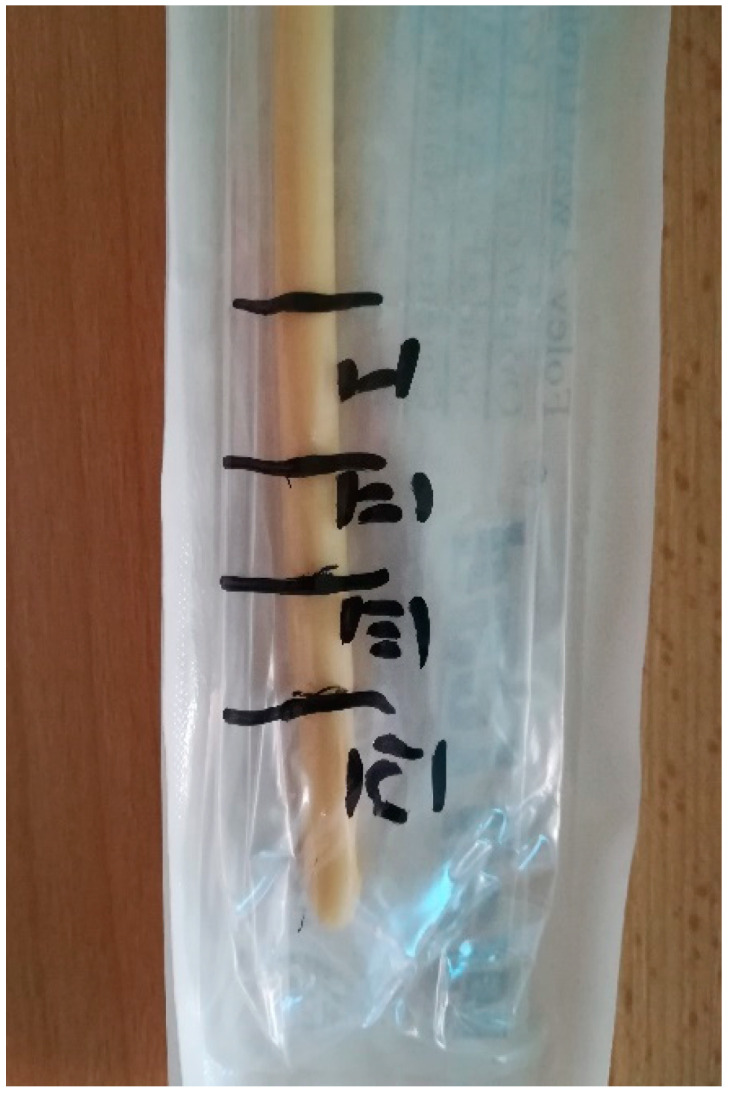
UC sampling.

**Figure 3 pathogens-10-00306-f003:**
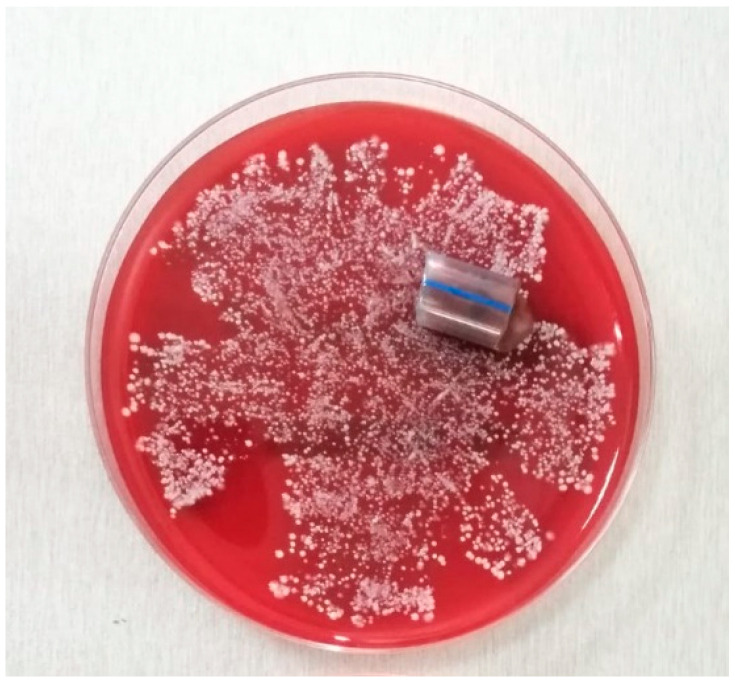
ET sample on culture medium.

**Figure 4 pathogens-10-00306-f004:**
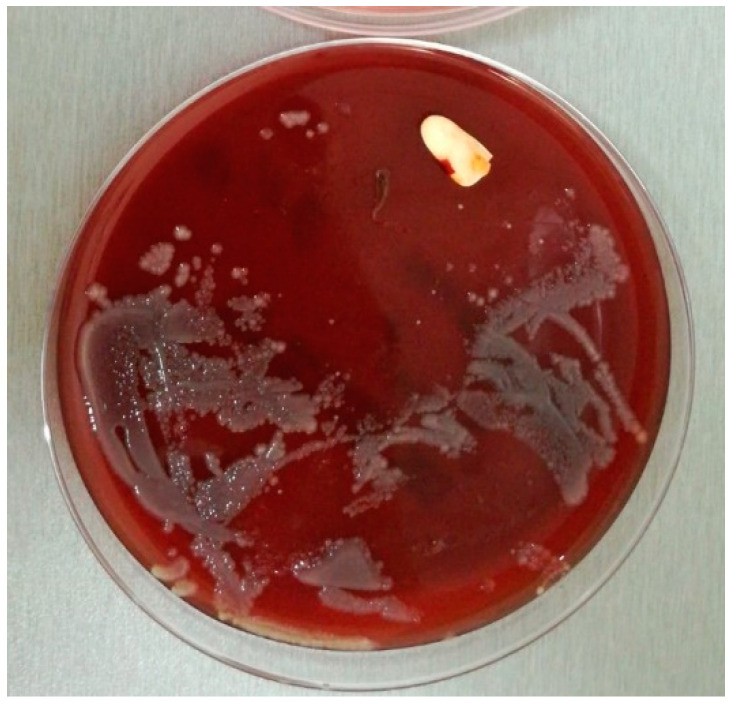
UC sample on culture medium.

## Data Availability

Not applicable.

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
