# Peer review of "Indwelling Device-Associated Biofilms in Critically Ill Cancer Patients—Study Protocol"

_pathogens, 2021, doi:10.3390/pathogens10030306_

Round 1
Reviewer 1 Report
Thanks for your efforts to accomplish this study.
Besides, I would suggest the following for the authors:
1- In the introduction at mentioning the quorum sensing (QS)and the less susceptibility of cells in biofilms to antimicrobials, I am afraid that these sections are mentioned in a näive way. May you please add more in-depth and detailed information on the QS, resistance, and tolerance mechanisms of biofilms to be understood by the reader, using recent authoritative citations?
2- In the introduction, may the authors elaborate on the acceptably high mortalities of septic patients (e.g., in numbers, places, and underlying reasons)? May the authors give examples of the most implicated bacterial biofilms in these cases (e.g., S. aureus in biofilms?) and dive deeply into it.
3- In the introduction, may the authors explain the time-consuming cell culture growth in terms of time and bacterial species to explicitly sharpen the meaning?
4- In the introduction, when the authors cited reference (20) for mentioning half of the patients with false-positive blood cultures, could they extensively say the context of these cultures and types of infections because this is a strong statement. May, the authors, mention specific studies as well by specific numbers and bacterial species to give a more profound highlight of the problem addressed?
5- In the introduction, may the authors explain scientifically what does critically ill means? Perhaps it could also be removed from the title.
6- Please, remove the statement in the discussion highlighting the publication and the presentation of results in an indexed journal and conferences to avoid any unintentional self-advertising. Instead, please focus on highlighting the concrete messages from this protocol scientifically and in comparison to other protocols and recent studies.
7- I would suggest that the authors change their aim of this study from being a protocol to an opinion or report, for example, because of the limitations (single-center study, approaching medical practice level, lack of control group).
8- In point 4.11., please substitute the originality sentence with explaining what this originality means in the study's real context? Please, again, avoid unintentional self-advertising.
9- I would urge the authors to add a control group to the study to have reliable results compared to the control.
Author Response
Response to reviewer 1
We are grateful for the time and effort you took to review our manuscript and we thank you for the constructive comments on how to improve its quality. Here are the improvements we made to the manuscript according to your valuable suggestions:
1,2,3 - The “Introduction” paragraphs discussing about “quorum sensing”, resistance and tolerance mechanisms of biofilms, about high mortality of septic patients and about the time needed for culture growth have been revised and updated according to the reviewer’s suggestions.
4- We clarified the paragraph linked to citation [20] from the “Introduction”, regarding antibiotic treatment of patients with contaminated or false-positive blood cultures. The cited authors refer to defensive medicine, because of excessive use of vancomycin to treat patients presenting with positive blood cultures because of contaminants.
5- For more clarity, we characterized the term “critically ill” in the Introduction.
6- We revised and improved the message of the scientific significance of our study in the “Discussion” paragraph.
7- Since the study is still ongoing we are considering adding a non-septic case-control group of patients, all in accordance to our financial possibilities. Until then, we accept changing the aim of our study to a case report one if necessary.
8- We revised the originality paragraph, enhancing the idea that the results of our study may contribute to the understanding of biofilm-associated infections in critically ill oncological patients and to the evaluation of the Next Generation Sequencing’s diagnostic power in cancer patients.
9- Thank you for the valuable advice. The lack of a group control is one of the study’s limitation. The cohort size is limited by the financial burden caused by Next Generation Sequencing use. Hoping that our results will be of value, we will consider having a non-septic control group of cancer patients who benefit from the same multiple organ support and treatment in future studies on similar topics with a larger cohort, in order to enhance the scientific power of our results.
We also reevaluated the manuscript for language and style errors. Our paper was previously checked for language, grammar, punctuation, spelling and overall style by an authorized English language editing company. Please let us now if further checking is required.
Thank you!
Reviewer 2 Report
Dear Authors,
the study protocol that you have proposed is of significant relevance, and can give an important contribution for the the practical aspects of severe infections management.
My suggestions for improvement, are mainly due to organisational issues, meaning that the information flow could be presented in a more clear way to the reader (avoiding for instance, the repetition of the same information in different points, and perhaps alerting better the reader that this is a Study protocol, so a heading named results, will not actually have results).
The current manuscript is divided in introduction, results, discussion and methods. My advice is to expand and divide the information presented at the materials and methods section to contextualise better the introduction and results (not sure if results is the best suited name for here, but didn't found journal guidelines about the specificities of a study protocol, I suggest simply call it Study Aims).
My suggestions include:
1) Introduce in your current introduction the general information provided in the discussion section between lines 151-168;
2) In the results section it would be nice to know the information explained at lines 284-286 (I do not believe this is common knowledge and would help the reader to understand better the study scheme without reading the full paper). NGS (line 133) is not written in full (and it is the first time it appears at the main text).
3) In the discussion section it could be useful some more information about what the Sepsis Indicating Quantifier score for circulating cell-free DNA consists on;
4) regarding the study design, I found no comments on how to address gender balance (although I understand that this might be very difficultto establish, a statement referring the thoughts given to it could be beneficial);
5) regarding the conventional microbiological culture, I have realise that one single temperature will be used (even for fungi growth), do the authors care to comment;
Best regards
Author Response
Response to reviewer 2
Thank you for the review and generous comments towards our study.
1, 2) We restructured our manuscript according to your suggestions. Also, in order to better highlight the aim of our study and not to confuse the reader, we changed the heading “Results” with “Expected results”, since our manuscript is a study protocol.
3) As suggested, we added for the readers more information about the Sepsis Indicating Quantifier score.
4) The investigators will enroll all consecutive surgical and non-surgical critically ill cancer patients admitted to the ICU, who meet all the inclusion criteria and do not comply with any of the exclusion criteria. Patient’s gender will be taken into consideration when analyzing the data, since gender specific differences exists in the incidence and mortality of certain cancers.
5) The microbiological laboratory of the hospital where the study is implemented can provide one constant temperature controlled environment for culture growth. To our knowledge, the majority of human pathogens optimum incubation temperature is 37 °C (including for pathogenic yeasts), although this might increase the ratio of bacterial to fungal growth rate. Because the majority of commensal yeasts can grow over a large range of temperatures, between 25 to 30° C, but only pathogenic ones withstand 37° C, we agreed to cultivate only in this condition.
We also reevaluated the manuscript for language and style errors. Our paper was previously checked for language, grammar, punctuation, spelling and overall style by an authorized English language editing company. Please let us now if further checking is required.
Thank you!
Reviewer 3 Report
The study protocol centres on comparing two methods (NGS and conventional culture based method) of pathogen detection/identification responsible for indwelling device biofilm formation in critically ill cancer patients.
It is not clear to me if the implementation of the study protocol at the Regional Institute of Oncology Iasi is ongoing and if so when did it start. Please can the authors clarify this.
The paper is well written and the different aspects eloquently explained. Hopefully the results from this study will allow us to understand, detect, control and hopefully stop biofilm formation on the indwelling devices of the patients.
The authors write “The results of the present study will be submitted to an ISI indexed scientific journal.” I am not used to reviewing manuscripts that only describe the “experimental methods” without presenting results and conclusion of the study.
A minor point: Temperature value and units separated by a space: 37 ºC.
Author Response
Response to reviewer 3
Thank you for your review and comments. We appreciate your feedback and thank you for taking the time to asses our manuscript.
The implementation of the study protocol started since 06.2019 and is still ongoing.
We also reevaluated the manuscript for language and style errors. Our paper was previously checked for language, grammar, punctuation, spelling and overall style by an authorized English language editing company. Please let us now if further checking is required.
Thank you!
Round 2
Reviewer 1 Report
Thanks to the authors for the revisions provided.
Please:
1- Do change this article to a case report.
2- Please, elaborate in more scientific in-depth details on the quorum sensing in the introduction with recent citations.
3- Please, remove from the conclusions the part that the study's design is the strongest point.
Thanks and best wishes!
Author Response
Thank you for your review and comments. We appreciate your feedback and thank you for taking the time to reassess our manuscript.
1- Our manuscript describes the protocol of an ongoing study that did not complete participant recruitment at the time of submission. It does not describe or interpret an individual case/case series, since all the characteristics of the cohort and the results of the indwelling device’s processing have not yet been obtained or interpreted.
2- We revised and updated the paragraph in the “Introduction” referring to “quorum sensing”, by adding additional scientific information (lines 92-107) and new recent references (10 and 11), all highlighted by Track Changes..
3- We excluded from the “Strengths and limitations of the study protocol” paragraph the part that the study's design is the strongest point (line 488).
We also reevaluated the manuscript for language and style errors and made a few style and spelling changes.
Thank you for all your help and suggestions!